# Political or environmental refugees? Re-examining the flight of the Vietnamese boat people, 1975-1995

Dr Saphia Fleury*

Wilberforce Institute, University of Hull
* saphia.fleury@hull.ac.uk

Thursday, March 13, 25

## Abstract

**This paper examines ecocide in South Vietnam during the Vietnam War (1954 to 1975) and the extent to which it drove the 'boat people' mass migration from Vietnam from 1975 to the mid-1990s. The exodus has become an archetype of a 'political' refugee flow, based on a Western Cold War narrative of people fleeing *en masse* from the persecution of an autocratic Communist regime. This paper challenges this assumption, showing how the interplay of environmental factors with political and military decisions contributed to the post-war exodus. These findings are reached through the analysis of historical primary sources as well as 229 oral history interviews, some 40 per cent of which were conducted with former child refugees. The implications have contemporary relevance because modern migration flows are frequently mixed, and climate change is further complicating the reasons that people leave their homes and their ability to access asylum. The conclusion argues that ecocide can, in some contexts, be considered a form of persecution for the purpose of refugee determination.**

## 1. Introduction

"*They make a desert, and call it peace*" – Tacitus

### 1.1 The argument

This paper re-examines the drivers of the 'boat people' exodus from Vietnam following the end of the Vietnam War. It begins with the somewhat controversial hypothesis that environmental factors, beginning with a deliberate military policy of ecocide, played a significant role – alongside political and economic factors – in driving people to leave Vietnam between 1975 and the mid-1990s. It is controversial since the accepted and overriding narrative of the boat people crisis, formulated in the West during the Cold War era, is one of Communist persecution of political opponents, removed from any environmental context.

I contend that environmental degradation and associated migration took four main forms during the period under discussion. First, during the war, millions of South Vietnamese peasants became internally displaced persons (IDPs) because of systematic wartime degradation of the environment ('ecocide'), perpetrated for military and political ends by South Vietnam and her allies – particularly the USA. This internal dislocation, and the inability or unwillingness of peasants to return to degraded lands following the war, prompted many IDPs to leave Vietnam altogether once the option arose after 1975.

Second, post-war food shortages directly caused by the wartime ecocide were a factor in some people's flight from Vietnam.

Third, the post-1975 government of reunified Vietnam used forced labour and forced relocation both to instil fear and control over the population, but also to re-establish the agricultural sector, increase crop yields and reverse the wartime environmental destruction. The realities of these systems, or even the threat of being subjected to them, prompted many Vietnamese, particularly the middle-classes, to leave.

Fourth, a series of extreme weather events in the 1970s – including floods, storms and droughts – further reduced the standard of living, including availability of food and arable land, particularly for the most vulnerable communities including IDPs. Landscapes that had previously been degraded by ecocidal weapons were especially vulnerable to the effects of extreme weather. These weather events in turn prompted the post-war Vietnamese authorities to increase their use of forced labour and forced relocation, driving people into ever-more intolerable conditions.

Sources connecting environmental destruction with overseas migration date almost exclusively from the crisis years (e.g., Grant, 1979; Desbarats, 1987) whereas twenty-first century literature focuses on the Vietnamese state's persecution of perceived political opponents as the sole driver of the refugee exodus (e.g., Kumin, 2008; Cadzow et al., 2010; Lipman, 2020). Between 1975 and 1995, hundreds of thousands of Vietnamese people were accepted as refugees by the USA, Australia, various European countries and other nations. These nations built a narrative to justify their welcoming of large numbers of refugees that also fed the Western Cold War anti-Communist discourse, based on the myth that the boat people were all political opponents of the Communist regime. Moreover, this narrative conveniently hid the reality that the USA and her allies had contributed to the refugee flow through their sustained campaign to destroy the Vietnamese countryside and agricultural sector. By the time the refugee flows had largely ceased in the mid-1990s, this had become the overriding narrative and continues to dominate the literature today.

My research uses oral histories and archival sources to show that environmental factors influenced the reunification government's abusive social policies in the post-war years, and that these policies contributed to the boat people exodus. Moreover, I show that some families directly attributed their migration to environmental destruction which caused loss of land and livelihoods, framing the exodus in more nuanced terms as a mixed-migration flow with multiple drivers including ecocide. This reframing can guide our understanding of modern-day migrations in which people leave or are displaced by multiple intersecting factors, related both to external drivers including the environment, climate change and state policies, and intersecting personal characteristics such as age, poverty and gender.

This reframing is partly achieved through the deliberate inclusion of child refugees' testimonies, which have been hitherto overlooked in many studies of the boat people. Children's perspectives and narratives, this paper argues, are less 'political' in that they have a more detailed, quotidian quality rather than relaying a macro-picture of the socio-political context for migration; a quality of some adults' testimonies. The importance of children's narratives to the present argument is further discussed below.

The fact that environmental factors – including extreme weather events such as flooding – did not happen in a political vacuum gives the findings further relevance to contemporary environmental migration, particularly debates around the applicability of the 1951 Refugee Convention to those fleeing environmental harms including the effects of climate change. This paper challenges the assumption that the defining clause of the Refugee Convention excludes people migrating for environmental reasons. The Convention defines a refugee as somebody who: "owing to well-founded fear of being persecuted for reasons of race, religion, nationality, membership of a particular social group or political opinion, is outside the country of his nationality" (UN, 1954: Article 1(a)(2)). Kent & Behrman (2018) argue that "a person displaced due to a flood or a hurricane cannot claim to be discriminated against or persecuted by any state, group or individual" (p.20). I dispute this for two reasons. Firstly, there are intentional, ecocidal actions that may intensify the impact of said flood or hurricane

(such as the deliberate US-led deforestation in South Vietnam, described below). Secondly, as argued below, state policies make certain groups more vulnerable to weather events, or indeed deem them legitimate targets of ecocidal actions and related persecution. This paper argues therefore that environmental harm can, in some contexts, be a form of persecution.

## 1.2 Scope and terminology

*The Vietnam War*

The conflict referred to in the West as the Vietnam War describes a period of international armed conflict following the withdrawal of French colonial powers from the region formerly known as Indochina in 1954 until the fall of Saigon – the South Vietnamese capital – to North Vietnamese forces in 1975. During this period and until reunification in 1976, Vietnam was divided, with a Communist government in North Vietnam, and a US-supported government in South Vietnam. Building on its earlier military and financial support for South Vietnam, between 1965 and 1973 the USA staged a major military intervention in both countries, supported by allies including Australia. The period of interest here commences with the widespread introduction of ecocidal weapons to the conflict in 1961.

The Vietnam War was fought across a wide geographical theatre, encompassing North and South Vietnam and neighbouring countries. This lengthy and complex conflict involved numerous state and non-state parties and varied military and guerrilla tactics across different terrains. The heavy bombing of North Vietnam and use of herbicides in Laos, for example, also wrought environmental damage. This paper focuses on South Vietnam, which suffered a deliberate miliary strategy of widespread environmental and agricultural degradation wrought by conventional and non-conventional weapons. Post-war agricultural reform also focused on the south.

*Boat people*

The term 'boat people' is used in a non-derogatory way to describe up to 2 million people who fled reunified Vietnam following the fall of Saigon 1975. The term is used by many of the former refugees themselves, often in a wider sense to encompass people whose journeys included land, sea and air travel. Some boat people also fled neighbouring Laos and Cambodia (Kampuchea). The boat people crisis concluded around 1995, when most of the refugees remaining in Asian transit camps were repatriated to Vietnam.

Data disaggregated by age is hard to come by, due in part to the high death rate among the boat people. However, it is estimated that around half of the refugees were under 18 at the time of departure. This paper attempts to give equal weighting to the perspectives and experiences of both child and adult refugees.

*Ecocide*

The word 'ecocide' was coined by biologist Arthur Galston to describe the destruction of South Vietnam's ecology during the Vietnam War, likening it to a crime against humanity (Weisberg, 1970). Such is the profound impact of ecocide on human systems that Gauger et al. (2012) have described it as a "crime against peace". This paper uses the legally contested term 'ecocide' as a shorthand for the deliberate military destruction of South Vietnam's ecology and agricultural sector. I also apply the broader definition of ecocide that includes both natural processes and anthropogenic degradation (White & Heckenberg, 2014), insomuch as I discuss the impacts of extreme weather events that struck Vietnam in the 1970s, which combined with wartime degradation to contribute to out-migration (see below).

Ecocide caused extensive damage to the landscape and economy of South Vietnam and the livelihoods of its citizens. Environmental degradation caused by conventional and non-conventional

weapons was documented extensively during and after the war by numerous scholars, scientific organizations and campaigners, notably Russell (1967); Bodenheimer & Roth (1970); Somerville (1970); Long (1970); Westing (1971 & 1983); Cairns (1976); Hickey (1993); Institute of Medicine (1994); Kerkvliet & Porter (1995); Stellman et al. (2003); Zierler (2011) and Martini (2012). Likely influenced by the legacy of its wartime experience, in 1990 Vietnam became the first country to include the crime of ecocide in its penal code (Gauger et al., 2012), highlighting the continued relevance of applying the term to the Vietnamese context.

While ecocide remains a flexible concept with a disputed definition and contentious legal standing internationally (Gauger et al., 2012), it encompasses a range of harms that allows connections to be drawn between historical environmental destruction and future risks. Expressing his anxieties at the height of America's use of chemical defoliants in South Vietnam, Weisberg (1970) noted:

"By tampering with the rivers, streams, food chains and cultures of a region, we set in motion chains of events which may, in time, have profound global consequences" (p.12)

These words could be describing the effects of climate change today. The concept of ecocide provides a framework to investigate the transnational and long-lasting effects of such harms, allowing us to respond to Weisberg more than 50 years later and describe the actual consequences of the chain of destructive events he predicted. In this case, our focus is on the profound human consequences of ecocide as a contributing factor to a mass migration that spanned the globe.

## 2. Background

### 2.1. Ecocide in South Vietnam: An overview

In 1961, several years before major ground troop operations began, US President John F. Kennedy authorized the limited, military use of chemical defoliants (herbicides) in South Vietnam. A declassified memo from Kennedy's National Security Advisor shows how, from the outset, herbicides were approved for use in "food denial", as well as to defoliate areas of potential Viet Cong ambush, with acknowledgement that this would necessitate internal "resettlement" of civilians (Bundy, 1961). A military official in Kennedy's administration noted "it is possible to 'sanitize' an area with chemical weapons, with gases and sprays that destroy animal life and crops. We can create a no-man's land across which the guerrillas cannot move" (cited in Zierler, 2011:68). By 1962, chemical defoliants – of which Agent Orange was the most notorious – were "a regular part of military operations in support of South Vietnam"; including an operation to destroy 3,642 hectares of mangroves in the Ca Mau peninsula, which "succeeded in stripping almost every leaf from the plant" (Zierler, 2011:77).

By 1963, food denial was an established military strategy of the allied forces in South Vietnam, with civilian crops accidentally destroyed and deliberately targeted to prevent them falling into Viet Cong hands. Peasants attempting to claim compensation for destroyed crops faced "bureaucratic obstructions", and the South Vietnamese army conducted "psychological operations... to assure peasants that herbicides were harmful neither to them nor to their animals" (Zierler, 2011:80), despite mounting evidence to the contrary. This expanded upon the South Vietnamese government's policy, begun in the late 1950s, of moving peasants off their land and into contained "agrovilles" or "strategic hamlets", from which they could not provide support, particularly food, to Viet Cong fighters.

The chemical defoliants deployed after 1961 were based on herbicides used on US farmland and were developed by American companies including Dow Chemicals. Their widespread use in the USA allowed subsequent administrations to underplay their health risks, despite the higher concentrations and quantities used in South Vietnam. The National Academies of Sciences, Engineering, and Medicine (2018) found that concentrations of toxic dioxins in defoliants used in Vietnam were up to three orders of magnitude higher than the manufacturing standards for herbicides used in the USA and noted that "about 77 million liters [of herbicides] were applied" in Vietnam between 1961 and

1971 (p.30). Despite herbicides' destructive capabilities, the successive administrations of US presidents Kennedy, Johnson and Nixon denied that they were subject to the 1925 Geneva Protocol banning chemical and biological weapons (Zierler, 2011; Martini, 2012).

The defoliation programme was a huge success in military terms and thus became self-perpetuating. As each area of forest, mangrove or cropland was cleared, Viet Cong fighters moved on, necessitating further sprayings, which were conducted from planes, helicopters, riverboats, trucks and by hand (Institute of Medicine, 1994). Yet defoliants were not the only weapon used by South Vietnam and her allies to deliberately degrade the country's environment and farmland. Other conventional and non-conventional methods included setting or exacerbating forest fires (Martini, 2012), carpet bombing (Bodenheimer & Roth, 1970; Somerville, 1970; Cairns, 1976; Westing, 1983), bulldozing (Somerville, 1970; Westing, 1983) and napalm (Robert, 2016). By 1964, napalm had reportedly already been used against more than 1,400 villages (Russell, 1967). Alexander (2000) notes that use of defoliants and napalm in tropical regions can trigger a chain of environmental disruption in the form of landslides and increased sedimentation of water bodies. Indeed, the chemically degraded areas were particularly badly hit in the post-war years by storms and flooding. A further food denial tactic was the capture of civilian stores of harvested rice and its deliberate destruction by contamination, burning or dumping it into rivers (Mayer, 1970; van Zyl, 2017). Throughout the war, defoliation efforts rarely distinguished between civilian and military targets (Zierler, 2011) and civilians bore the brunt of agricultural and ecological degradation and food denial.

As the war dragged on, first the scientific community, then the public, and finally international observers increased their opposition to defoliants. Dow Chemicals' production of Agent Orange was equated with the manufacture of Zyklon B by IG Farben, for which the latter's directors were prosecuted during the Nuremberg Trials (Zierler, 2011). In 1969, UN General Assembly Resolution 2603 undermined the USA's reading of the 1925 Geneva Protocol as excluding the use of chemical compounds that were toxic to plants and, in 1971, the USA ended its decade-long defoliation programme in South Vietnam.

## 2.2 Push factors for the exodus: The dominant narrative

### 2.2.1 Twentieth versus twenty-first century literature

Twenty-first century literature on the Vietnam War and its aftermath focuses on the political drivers of the exodus, with scant mention of environmental or even economic drivers. For example, a UK A-Level history study guide (Sanders, 2007) mentions Agent Orange only in the context of US veterans' health, with no acknowledgement of its environmental impact. Max Hastings' well-regarded 2018 account of the Vietnam War runs to 700+ pages but devotes little more than one page to the use of herbicides and omits any mention of environmental drivers for migration (Hastings, 2018). Similarly, Hall (2018) skims over the 10-year defoliation and food-denial campaign in just three sentences and attributes the boat people exodus to "social dislocation" caused by economic collectivization (p.90), without describing the environmental context in which those economic policies were designed.

Cadzow et al. (2010) provide a typical example of recent narratives on the boat people's motivations for fleeing, which focus on political and ethnic drivers:

"Life became increasingly difficult for South Vietnamese government associated people and for Vietnamese with Chinese ancestry. They began to leave after the socialist government closed private businesses in 1979." (p.116)

Other recent sources (e.g., Kumin, 2008; Lipman, 2020) also cite the political persecution of the ethnic Chinese Hoa population, and no doubt there was increasing pressure on Hoa people to leave the country, particularly as tensions flared along the Vietnam-China border. Yet earlier accounts by Kushner & Knox (1999), Desbarats (1987) and Grant (1979) all note that, like the ethnic Vietnamese population, the Hoa were also victims of the New Economic Zone (NEZ) forced internal resettlement

programme and other post-war agricultural policies: "Most Chinese in Vietnam did not want to be sent to the countryside" (Grant, 1979:87).

While the narrative in resettlement countries until the early 1990s was that refugees were fleeing 'Communism', a survey of Vietnamese refugees in Britain found that only 4% cited the political system as their reason for leaving (cited in Kushner & Knox, 1999). More commonly cited motivations were internal displacement resulting from the NEZ system, fear of forced labour in re-education camps, and displacement due to extreme weather events. Nevertheless, the study also found that many professional people had left because of the loss of money, property or position. One refugee said he had used all his savings buying food on the black market and was left with just three pounds of rice per week to feed his family. This example shows how the intersection of military environmental degradation, extreme weather and agricultural and economic policies became drivers of individual migration decisions.

While acknowledging other push factors, Tsamenyi (1980) highlighted the devastation wrought by a series of typhoons and floods in 1978, which contributed to starvation and the environmental degradation of the south:

"It is likely then that the impact of these natural disasters also contributed to the exodus of people from Vietnam. This argument is supported by interviews conducted among boat people, some of whom referred to food shortages as a major reason for leaving Vietnam." (Tsamenyi, 1980:7)

Speaking in 1978, Swedish statesman Hans Blix also attributed the exodus to "several periods of natural catastrophes" resulting in food shortages (cited in Grant, 1979:97). This was echoed by a refugee interviewed by Grant in Australia, who stated that "most Vietnamese left because of food shortages" (p.182). The peak years of the boat people departures – 1978-1980 – indeed coincided with extreme weather events. Yet, the characterization of extreme weather as "natural disasters/catastrophes" by post-war commentators overlooks the fact that typhoons and floods were falling on land already severely degraded by deliberate, wartime deforestation and were hitting displaced and vulnerable populations. Both natural and human resilience to these weather hazards had therefore been depleted by military tactics, political policies and social conditions, making their effects more significant (see also 4.1 below).

Under the NEZ system, which was established in 1976 by the newly reunified Vietnamese state, urban families were assigned often remote and unproductive plots of land to farm, sometimes resulting in starvation, and were cruelly punished if they attempted to return to the city. The government set a target to relocate 4 million people to NEZs by 1980 (Thrift & Forbes, 1986), and around one-fifth of the total population by the end of the century (Desbarats, 1987). The early days of the relocation programme involved reinstating peasants on land they had been forced to abandon during the war. However, often their villages no longer existed, and many IDPs were reluctant to return or had no land to go to. When too few people volunteered to move to NEZs, efforts to force resettlement ramped up, particularly targeting IDPs, the unemployed, small traders and people seen to pose a political threat (Desbarats, 1987; Dalglish, 1989; Bradley, 2009).

Desbarats (1987) records how refugees in Australia described severe malnourishment in the NEZs, in part because of a lack of farming skills among urban-dwellers forcibly relocated to the countryside, and subsequent high death rates, particularly among children. Such connections between the environment, agriculture and migration are largely missing from literature published after the repatriation of Vietnamese so-called 'economic migrants' in the mid-1990s. The change in political attitudes towards the boat people is thus mirrored in the focus of subsequent literature. The key exceptions are those studies that directly interviewed refugees in camps, such as Freeman and Huu (published 2003, but conducted in the 1990s) who recorded themes of agricultural policy, re-education camps and the NEZ system in refugee's narratives. Once researchers stopped speaking directly to refugees about their experiences, environmental concerns disappeared from the literature completely.

The high level of internal displacement during and after the Vietnam War, driven in part by environmental damage, provided a motivation for onwards migration once opportunities arose after 1975. Literature from the war era describes poor conditions in camps for IDPs:

"[The camps] are placed in the baking sun on bulldozed earth lots surrounded by barbed wire.... A refugee, or 'detainee', is left without any reason to live, frequently separated even from friends and family in the evacuation shuffle. During 1967 and 1968... the resettlement camps were unable to provide even potable water, food and shelter, much less medical aid, clothing and a new life." (Schell & Weisberg, 1970:26)

French philosopher Jean-Paul Sartre (1970) described the internally displaced Vietnamese peasants as worse off than slaves, "reduced to a living heap of vegetable existence" (p.41). In 1979, Grant noted that some people had been displaced several times after successive attempts by the government to resettle them in rural locations. In 1989, Dalglish argued that internal displacement caused by "war[time] devastation" of the countryside was one driver of the exodus (p.18). Returning home to severely degraded lands and destroyed villages was often not an option. Cut off from their former communities, livelihoods and in some cases family members, many IDPs felt they had nothing to lose in attempting to flee Vietnam.

The post-war internal dispersal of millions of people, including many already displaced by war and famine, was driven by an imperative to feed the growing population and reclaim 5 million hectares of degraded arable land and 7 million hectares of deforested hillsides (Grant, 1979). These relocations did not become voluntary until 1991 (Anh & Huan, 1995). British philosopher Bertrand Russell's renowned 1967 book, *War Crimes in Vietnam,* focuses heavily on the effects of defoliants and napalm on the civilian population, as did his influential letters to the press throughout the 1960s. Yet these narratives of internal displacement, relocation and environmental destruction are underrepresented in recent re-tellings of the Vietnam War. My findings in section 4 below resurrect this history and demonstrate the extent to which these factors drove emigration post-1975.

### *2.2.2 Child migration drivers*

The drivers of children's migration[1] are often assumed to be the same as for their parents, to the extent that children's experiences are frequently excluded from migration literature altogether (Bhabha & Young, 1999; Orellana et al., 2001; Thronson, 2018). Christensen & James (2017) found that, "[t]raditionally, childhood and children's lives were explored solely through the views and understandings of their adult caretakers who claim to speak for them" (p.4). However, the push factors for child migration are often quite distinct and, in this case, are important to record given the high numbers of unaccompanied Vietnamese minors – approximately 60,000 – who arrived on foreign shores post-1975, and the fact that an estimated 50% of the overall Vietnamese refugee population were children (Grant, 1979; Freeman and Huu, 2003).

While Dalglish's 1989 study *Refugees from Vietnam* tends to overlook the experiences of children, one story stands out as highlighting the desperation of lone and abandoned minors to flee the country:

"My parents were then sent to prison, my elder brothers and sisters were sent to a place far from home to work as slaves... leaving my younger brothers and sisters and myself, who were all under twelve, behind... [The authorities] put us [children] in a cottage near a forest and gave us food that even a dog would not eat. All we could do was cry." (Cited in Dalglish, 1989:20)

In this case, the young siblings fled Vietnam using their own initiative and resources. In other cases, lone children were sent out of the country by their parents. Some children departed in family groups and became separated or lost their parent(s) at sea. Still others, such as the 10-year-old boy interviewed in the example below, accidentally became unaccompanied refugees:

"One night, Hai and his friend decided to sleep on the roof of a boat owned by his friend's father. They fell asleep looking at the stars. They awoke the next morning when thirty-two people climbed aboard… Hai said he wanted to swim to shore, but a man said, 'If you jump overboard, I'll shoot you.' Hai recalls, 'I felt terrible. I missed Mom. I was crying. I didn't know where we were going, and I was panicked.' After six days, they landed in the Philippines, and after six months he was brought to the United States." (Freeman and Huu, 2003:157)

Freeman and Huu's study (undertaken in the 1990s) is an important exception to the focus on adults' narratives in the Vietnamese refugee literature. Moreover, they repeatedly record themes of agricultural work in child refugee's narratives:

"After my father came back from reeducation camp, he became a farmer, the only job he was allowed to hold…. I had to work all day to help support the family". (Unaccompanied refugee minor, cited in Freeman and Huu, 2003:142)

Wherever possible, the findings in section 4 include children's narratives alongside those of adults to ensure that their unique experiences of environmental drivers for leaving Vietnam are represented.

### 2.2.3 The journey and resettlement

Understanding the connections between environmental destruction, internal displacement and emigration is key to understanding the boat people's experiences during and after their journeys. Under the 1951 Refugee Convention, reasons for leaving one's country subsequently determine one's legal status, and thus eligibility for protection and support. Vietnamese refugees initially existed in legal limbo and, although large numbers were subsequently granted asylum abroad, their legal status was continually disputed, often leading to grave human rights abuses and even death. This prompted Kumin (2008) to conclude that: "Not since the Second World War had the international community witnessed the denial of asylum so vividly and dramatically" as in the boat people crisis (p.106). This situation was occasioned by a combination of the large numbers seeking protection and a weakening desire among the international community to abide by their ever-increasing human rights obligations (Fleury, 2023).

The literature records several means of escape from Vietnam between 1975 and the early 1990s, which included leaving secretly and illegally with the risk of being imprisoned or killed if caught; leaving with government approval after paying large amounts of gold; bribing government officials to turn a blind eye; and complying with official state efforts to remove certain people from the country, including via an 'Orderly Departure Programme' (ODP) managed by UNHCR, the UN refugee agency (Grant, 1979; Kushner & Knox, 1999; Vo, 2006; Kumin, 2008).

The journeys themselves varied tremendously in terms of route, length and risk of harm. People departing by boat might be betrayed to the authorities or shot by Vietnamese coastguards. Depending on the time of year and route taken, boats might encounter typhoons, gales, baking sun or monsoon rain. Pirate attacks were common; in June 1979, US officials estimated that 30% of boat people leaving southern Vietnam had been victims of rape, pillage or murder by pirates (Grant, 1979). Boats in distress might be ignored by passing vessels or towed away from the coastlines of neighbouring countries to prevent disembarkation. In many cases, children set out unaccompanied, either of their own volition, or because their families could not afford to accompany them, or to spread the risk of losing the whole family in one disaster. Hundreds of thousands of people drowned or died of hunger, thirst or exposure while drifting at sea (Vo, 2006).

Those who made it to Vietnam's neighbours – Malaysia, Indonesia, Hong Kong, Singapore, the Philippines or Thailand – were usually detained in transit camps and processing centres established by the local authorities and/or UNHCR, where they faced "boredom, illness, anxiety and restriction" (Cadzow et al., 2010:123). Although individual camps varied in terms of resources and security (Vo, 2006), they were frequently overcrowded, dangerous and lacking in basic facilities (Kushner & Knox, 1999; Lipman, 2020).

In the USA, Vietnamese arrivals were hailed as "model refugees: hard-working, well motivated and eager for self-sufficiency" (p.161); nevertheless, racial tensions were ignited over access to housing and jobs, and many suffered residual trauma. In Australia too, tensions arose over employment opportunities, while in Malaysia, Vietnamese boats were stoned from the shore by locals who argued that refugees drove up the cost of living and drained government resources (Grant, 1979).

In the UK, emotional difficulties arose in a traumatised refugee population. Refugees faced hostility from some quarters (Crangle, 2016), although volunteers did their best to help them settle (Kushner & Knox, 1999). As with children's experiences of departure, the impact on children's lives is largely absent from the literature of resettlement. Each of the three UK government-commissioned studies on the Vietnamese refugee community (Jones, 1982; Edholm, 1983; Duke & Marshall, 1995) fails to mention children's needs. This may have contributed to the fact that some children failed to thrive in school, and rates of domestic violence were high (Kushner & Knox, 1999).

Although the number of Vietnamese refugees eventually resettled in the UK was small compared to the USA – approximately 25,000 compared to more than 1 million[2] – they were from more diverse backgrounds. Many refugees resettled in the UK had a peasant or subsistence fishing background, including some of the Hoa (ethnically Chinese) refugees (Kushner & Knox, 1999). This finding partly contradicts the narrative that the Hoa were middle class with political motivations for migrating. UK-bound refugees also included "a high number of unaccompanied children, mainly those picked up at sea [and for whom] relatives could not always be traced" (Kushner & Knox, 1999:326). Robinson (1989) noted that Britain was less selective in choosing whom to resettle than other destination countries, perhaps explaining the lack of a political or persecution-based motivation for departure in the narratives of UK-based refugees and the higher proportion of peasants and unaccompanied minors among those accepted by the UK. This profile suggests that environment-related concerns may have been a greater contributor to migration decisions for refugees resettled in Britain compared to elsewhere.

## 2.3 Moving the debate forwards

The literature demonstrates the catastrophic effect that both ecocidal weapons and resultant post-war land reclamation policies had on the southern Vietnamese countryside and its inhabitants. Further, it partially demonstrates how the ecocide became a contributing factor for internal displacement and the subsequent boat people migration. This narrative is in evidence in wartime and immediate post-war sources but disappears from the literature after around 1995, in favour of political causes of migration. Further, the literature shows how the international response was mixed, with some boat people gaining refugee status and a warm welcome, and others being refused protection and ultimately repatriated. The literature does not shed light on the role of environmental migration drivers in such responses. My research findings outlined below expand our understanding of the role of ecocide in shaping the post-1975 exodus. Based on this expanded understanding of the role of environmental migration drivers, I make the argument for refugee determination procedures taking ecocide into consideration as a form of persecution when perpetrated against specific groups.

# 3. Methodology

The research for this study was conducted between 2019 and 2022 as part of my PhD project at the University of Hull, using existing oral history collections and archival documents. A fuller explanation of the methodology and its ethical considerations can be found in Fleury (2023).

## 3.1 Oral histories

I examined a total of 229 oral histories of Vietnamese refugees in the UK and USA from four collections:

1. University of California Irvine, VietStories oral history project: 158 transcripts from Vietnamese refugees in the USA.
2. British Library / Refugee Action Vietnamese Oral History Project: 32 oral histories from Vietnamese refugees in the UK.
3. Vietnamese Boat People Podcast: 27 episodes were analysed, detailing the stories of 20 refugees in the USA and several practitioners who supported them.
4. Voices of the Vietnamese Boat People: 19 narratives were analysed based on interviews conducted with refugees in the USA.

I pseudonymised the oral histories by assigning each narrator a random three-digit number preceded by two letters denoting the collection from which they came:
CI = University of California Irvine
BL = British Library/Refugee Action
VB = The Vietnamese Boat People Podcast
VV = Voices of the Vietnamese Boat People

Through these oral histories, I investigated the importance of the environment, agriculture and related themes to people who migrated from Vietnam to the UK and USA between 1975 and the mid-1990s.

As discussed above, I included where possible the oral histories of former child refugees. Children are an overlooked group in refugee studies (White et al., 2011; Singleton, 2018; Black, 2019) and, as explained in section 2.2.2 above, I found this to be largely true for studies of Vietnamese refugees, even though around half were children. I included children's voices, especially voices of unaccompanied minors, to help redress this. At least 91 of the oral history participants (some 40 per cent) were former child refugees. (Age at the time of departure was not always given, so the real number may be higher.) Children's viewpoints on migration are sometimes criticized as being incomplete, since children may be excluded from family migration decision-making and may be less aware of their country or community's socio-political context. Yet, in this case, I found that many children migrated independently of their families or had unique insights that were missing from adults' narratives. This is partly because children were not required to repeatedly retell their story in refugee determination procedures nor to shape their personal narratives to fit the expectations of such procedures. Thus, children were more likely to speak from their own experience rather than describe macro-level policies driving their situation. This makes children's contributions essential to building a fuller picture of experiences among the boat people.

**3.2 Historical sources**

I gained broader context by drawing on archival sources to investigate the importance of ecocide as a direct or indirect driver for the post-1975 exodus. These included personal correspondence, minutes of meetings, press releases, pamphlets, reports, government briefings and communications, newspaper articles and NGO records, among others. I also conducted an email exchange with Dr Jean-Pierre Guignard, the author of one archived document published in 1967, to understand the context for his study (see 4.1 below).

Documents relating to ecocide during the Vietnam War and proceedings of the post-1975 Vietnamese government came from the following repositories:

1. Southeast Asia Collection, University of Hull, UK.
2. Hansard records of UK parliamentary proceedings, 1960-1975.
3. National Archives and Records Administration, USA.
4. US Department of Agriculture, USA.

Documents relating to the boat people came from the following repositories:

1. Race Relations Resource Centre, Manchester Central Library, UK.

2. Hansard records of UK parliamentary proceedings, 1975-1998.
3. Prime Minister's Office Records for 1979, National Archives, Kew, UK.
4. Papers of the UK Joint Committee for Refugees from Vietnam (JCRV), 1979-1982, National Archives, Kew, UK.
5. Gerald R. Ford Presidential Library records, 1974-1975, USA.
6. Amnesty International Archives, 1975-1996, London, UK.
7. Hull History Centre, UK.

A total of 98 historical documents were analysed across these repositories.

### 3.3 Validity and relevance of the data

My investigation of this data takes the form of supra-analysis, in that my research questions transcend the purpose for which the data was originally collected (Heaton, 2008). I acknowledge that I will, in some cases, be reframing people's narratives to highlight my own research interests, for example by looking for reasons for an individual's economic situation in clues given about ecological destruction or agricultural practices in their immediate environs. Nevertheless, I aim to avoid reframing their histories in such a way that ignores their own concerns and memories.

Using secondary sources raises the question of 'data fit': whether the data is truly relevant to my study. In this case the data was collected for different purposes to my own research aims. However, I believe this to be a valuable attribute of the oral histories. In oral histories, people are asked to describe those aspects of their experience that matter most to them. In doing so, they may inadvertently repeat themes relevant to a specific area of research, which may at the time appear incidental. Historian Alessandro Portelli argues that the value of oral histories lies precisely in this selective retelling. Oral history interviews, he argues "always cast new light on unexplored areas of the daily life of the nonhegemonic classes... they tell us a good deal about [an event's] psychological costs." (Portelli, 2006:36). In my exploration of this data, I found the themes of agriculture, internal displacement and human rights violations coming up repeatedly, along with their "psychological costs", even though they are not necessarily central to the narratives and do not feature in the interviewers' lists of questions. This suggests that these themes shaped the storyteller's experiences and left a psychological imprint, regardless of whether they or the interviewer felt they were significant. I believe this repetition is more telling than if I had 'led' participants in a structured interview on the role of the environment in their displacement.

A lingering question remains: 'What is missing from these archives?' One oral history participant, who had herself been an interviewer for the project, commented that some people withdrew their testimonies for fear of a community backlash, stating: "definitely we did miss some of them" (CI124). Other oral histories remain closed to public access at the request of the interviewee. Unfortunately, the oral histories can never be fully representative of the Vietnamese migrant experience, not least because of the extremely high death toll among those who took to the seas. I acknowledge, therefore, that many boat people's stories will never be told.

## 4. Findings

I present my findings in four parts. Section 4.1 shows how ecocide was a deliberate form of political and military persecution against certain groups within the Vietnamese population perpetrated by South Vietnam and her allies, particularly the USA. Section 4.2 shows how the ecocide drove internal displacement and post-war overseas migration, sometimes in conjunction with extreme weather events. Section 4.3 shows how the post-reunification Vietnamese government responded to the wartime ecocide through programmes of forced labour and forced resettlement, which further persecuted particular groups and contributed to the boat people exodus. In each case, ecocide and its consequences were based in political decision-making which resulted in the persecution of certain groups and communities, ultimately driving many of the victims to leave the country. Based on this finding, and in conjunction with an analysis of the legal status of boat people, I argue in section 4.4

that environmentally linked persecution would have met the threshold for some individual victims to be considered refugees under the Refugee Convention definition.

## 4.1 Ecocide as persecution

Numerous archival documents attest to a growing concern around the military use of chemical agents, particularly defoliants, in Vietnam beginning in the mid-1960s. Swiss pharmacologist Dr Jean-Pierre Guignard compiled evidence relating to chemical and bacteriological warfare in South Vietnam and, in 1967, disseminated his findings in a document highlighting the US government's campaign of misinformation:

"Large numbers of doctors, chemists, bacteriologists and technicians are engaged in this work [producing chemical weapons]. But only 15 per cent of their studies are published in the scientific journals; the greater part of their work makes up a secret literature, the exclusive property of the U.S. Department of Defense" (Guignard, 1967:4)

Dr Guignard's publication demonstrated that, as early as 1967, the dangers to human health from defoliants were common knowledge, as was the US military's strategy of food denial via chemical warfare. Guignard (1967) compiled evidence of "extremely grave ailments", particularly in children and the elderly exposed to defoliants (p.5), and "[h]eavy crop destruction" beginning in 1963 (p.14). When I contacted the long-retired Dr Guignard in April 2021, he recalled that his publication had been part of a strategy by Swiss doctors to "denounce the plans of the Pentagon to develop chemical weapons".

Meanwhile, journalists and relief agency staff (e.g., Pepper, 1967; Carlisle, 1969) were recording injuries and fatalities among children exposed to defoliants, white phosphorus and napalm. In 1968, UK MP David Kerr noted that this "chemical and biological warfare… is, in fact, occasioning starvation among an already starving population". The UK Foreign Secretary responded that it was "inevitable in the operations of war that there will be an interdiction of food supplies" (Hansard HC Deb., 20 May 1968).

Section 2.1 above described how ecocide was a deliberate military tactic of the US and South Vietnamese militaries, with the twin aims of destroying crops and creating wastelands through which enemy combatants could not travel. Much of this was achieved using defoliants. The UK Government supported the use of defoliants by the USA and South Vietnam, despite mounting evidence of their harms, including that compiled by Dr Guignard which was widely circulated in several languages. In 1970, UK government minister George Thompson claimed: "We have no evidence that [defoliant] use in Vietnam is causing lasting harm to the ecology of the country or is having any poisonous effects on human beings" (Hansard HC Deb., 6 April 1970). The continued support for the US position from the UK Government was evidenced in 1972, when Foreign Office Minister Baroness Tweedsmuir was questioned about the destruction of crops, irrigation systems and soil by US bombing. Her reply: "My Lords, all wars involve destruction of the environment." She confirmed she had assurances that the USA "would not bomb dykes deliberately, although they could possibly do some incidental damage" (Hansard HL Deb., 28 July 1972). Such support by the USA's allies shows that persecuting sections of the Vietnamese population through environmental destruction was not only a military strategy; it had political support that went further than South Vietnam and the USA.

Moreover, the deliberate ecocide perpetrated in South Vietnam was targeted to persecute a certain politicized group: peasant farmers who might lend their support to the Viet Cong. In this way, the Refugee Convention definition begins to have relevance: the victims were "being persecuted for reasons of… membership of a particular social group or political opinion" (UN, 1954: Article 1(a)(2)). Within this group, the sub-groups of children and the elderly were particularly affected by chemical agents (Guignard, 1967) and food denial (Mayer, 1970).

## 4.2 Ecocide as a direct driver of displacement

In each of the four oral history collections analysed, issues surrounding agriculture and the environment were raised, albeit by a minority of refugees. For example, in the California Irvine collection, 19 out of 158 participants specifically discussed subjects such as the effects of bombing on the country-side, and pressures on those whose livelihoods depended on farming.

That a minority of refugees raise these themes in their oral histories is likely the result of a lack of interview questions on environmental and agricultural issues. The California Irvine project, which represents 69% of the oral histories analysed (158 out of 229 interviews), provided interviewers with 133 questions/discussion topics, only 14 of which relate to wartime and post-war experiences *within* Vietnam.  None of the questions enquire specifically about environmental or agricultural issues or ecological degradation. Likewise, the British Library/Refugee Action interviews (representing 14% of the oral histories analysed) did not include questions specifically on these topics. Interviewees who raised these subjects therefore did so spontaneously, despite the constraints of the interview, suggesting that they were noteworthy features of their pre-migration experience. It cannot be estimated how many other participants may have shared similar experiences had they been directly asked. However, I conclude that the incidence of people spontaneously mentioning ecocide-related themes, although low, is significant enough to suggest that they acted as migration drivers for some refugees. This is particularly true for issues relating to forced agricultural labour and forced relocation to NEZs (see 4.3 below); the environment-related themes most discussed across the four oral history collections. This section considers the extent to which ecocide drove internal and overseas displacement.

A wartime briefing produced by the US Navy's Scientific Advisory Group (Warren, 1968) presented significant evidence that chemical defoliation was causing starvation and internal displacement among the Montagnard Indigenous people in South Vietnam's Central Highlands. Extreme weather exacerbated internal displacement. The Communist Party Committee in one coastal province noted in 1974 that typhoons and drought which struck that year had been overcome. People internally displaced by the extreme weather were now self-sufficient, but "food supply in combat areas is not adequate", according to the Binh Dinh Province Party Committee (1974:31-32). While such committees might be inclined to exaggerate local successes, the statement nonetheless demonstrates that extreme weather was driving further internal displacement, and that officials were preoccupied with agriculture, natural disasters and food availability on the eve of peace.

In my email exchange with Dr Guignard in 2021, he directly linked the ecocide with human migration: "Destruction of the crops and insecurity clearly pushed the population to abandon their villages and farms" for cities, and subsequently some of these refugees "escaped from these crowded areas as boat people", he explained.

As the boat people refugee crisis peaked in the late 1970s and made headlines in Europe, it was noted in the UK parliament that many of the refugees were pastoralists or fisherfolk (Hansard HC Deb., 15 December 1978), not only political refugees as assumed by most twenty-first century accounts. A 1979 Oxfam report records several environment-related migration drivers, including:

"Above all... a sustained attempt... to manipulate – and completely transform – the entire physical and human environment of a country... The natural vegetation cover, the rainfall regime, the soil pattern, the natural drainage pattern, the complex of animal life, and the complex of disease-bearing organisms were all... manipulated to fight the enemy" (Ashworth, 1979)

The Oxfam report further describes the "infertile, cratered moonscapes" left by bombing raids, as well as defoliation, bulldozing, erosion and flooding. Similarly, a 1982 handbook to assist UK health professionals in supporting refugees mentions defoliation of crops and bombing of irrigation systems as relevant to the experiences of Vietnamese refugees, reminding health workers that "the country still has not recovered" from the wartime damage to food supplies (Mares, 1982:108).

Varied contemporary archival sources therefore support the hypothesis that ecocide, which exacerbated the impacts of subsequent extreme weather events, contributed to internal displacement and, ultimately, to people fleeing Vietnam. This is borne out by oral histories of refugees who resettled in Britain and the USA, some of whom directly mentioned environmental factors as colouring their pre-migration experience and migration decisions, including food shortages and extreme weather. For example:

"[T]he war came through [our village], and not a single structure survived. There's nothing survived. Even trees, we don't have any old trees there. Everything wiped out" (CI138, male child[3])

"[W]ar destroy the environment, destroy the people" (CI084, male adult)

"I would say the majority, at least 80%, the people in my neighbourhood did not have enough to eat everyday... the war happened. A lot of crops got destroyed." (CI065, male unaccompanied child)

"A lot of people were worried about food mostly." (CI056, male adult)

"We did not even have enough food to eat or survive by that time" (CI061, male adult)

"[There were] a lot of hurricanes and flooding, and those houses made of straw and leaves were blown away." (BL026, female adult)

These speakers came from a range of backgrounds; socially, economically and geographically. They also represent a range of ages. Yet they share common memories of ecocidal destruction and food shortages, attesting to the indiscriminate nature of this phenomenon across Vietnamese society. Nevertheless, as the next section shows, certain groups were more vulnerable to the impacts of these ecocidal patterns.

The sources cited above attest to both the vast extent of deliberate ecocide and the effect of storms, flooding and drought on South Vietnam's economy and populace. The effects of extreme weather events were multiplied by ecocidal military tactics and the vulnerability of the already displaced population. The far-reaching consequences included periods of famine, internal displacement and a harsh government programme of forced relocation, demonstrating that the environmental impact of defoliants, bombs and other ecocidal weapons in rural areas contributed to post-war primary and secondary migration.

## 4.3 The Vietnamese state's response to the ecocide

A month prior to the fall of Saigon in April 1975, US Congressman William Chappell informed President Ford that the South Vietnamese government was already taking IDPs from camps, putting them "through training programs" and dispersing them to "the countryside, fishing, etc., where they could be productive". This was considered a positive move to grow South Vietnam's agricultural sector, and Chappell called for US aid for these dispersal programmes (White House, 1975). These policies, designed to reverse the loss of agricultural capacity due to war damage, relied on forced internal relocation of already displaced families, not necessarily to their region of origin.

Meanwhile, wartime documents demonstrate anxiety among the North Vietnamese authorities over the destruction of agricultural land by a combination of ecocidal weapons and extreme weather in South Vietnam, as they anticipated reunification. An article in North Vietnamese newspaper Hanoi Hoc Tap (1972) notes that:

"[T]he fundamental role and strategic significance of agriculture have become more prominent under present circumstances. First of all it is necessary to develop the production of grains and foodstuffs in the [Mekong] delta, the middle region and the mountain region. It is necessary to urgently

complete the preparation for the struggle against floods and the calamities caused by nature or the enemy during the coming rainy season" (p.82).

In 1973, the North Vietnamese authorities waging war in South Vietnam (known to the USA as COSVN), issued a directive to regional committee members stating that they should focus on, among other things, "urging the people to return to their ricefields and orchards" through propaganda activities and slogans (COSVN, 1973:10). COSVN's political strand, the People's Revolutionary Party, noted in its economic plan for the envisaged post-war era:

"The rural area [of South Vietnam] has been heavily devastated by war.... Waste land is immense. Irrigation and flood control systems have been almost entirely destroyed.... The peasants' reserve stock is nil. It is too late for the people returning recently from enemy-controlled areas to work for the main crops. Flood disaster will cause us difficulties in stabilizing the people's life and production activities. In some places, famine, disease, and shortage of salt are occurring" (People's Revolutionary Party, 1973:10)

Key challenges envisaged by the Party included: "motivat[ing] rural people who have been relocated by the enemy to return home to farm" (p.13). The document emphasizes the risk of famine if the ecocide is not quickly reversed.

Such sources are evidence of the context for political decision-making within Vietnam as the war ended. Ultimately, South Vietnamese officials, and later the government of reunified Vietnam, used this context to justify a mass programme of relocation and persecution, including forced labour. This persecution was politically driven but motivated in large part by the perceived need to rehabilitate the degraded countryside.

Archives show that the environmental and human impacts of ecocide in Vietnam continued for decades after cessation of the conflict. In the mid-1980s, agricultural production remained the leading concern of reunified Vietnam, with forced relocation the government's main solution. In a pamphlet from 1984, Communist Party General Secretary Le Duan notes the imperative to restore damaged forests and cropland, so that "every hectare of land is exploited". Restoration of land through population relocation would be approached in a military fashion: "the work force must be redistributed on a nation-wide scale to expand farming areas with the same zeal as in the fight against the enemy" (Duan, 1984:10).

Those forcibly relocated to the countryside, which the UK's Daily Telegraph (1979) likened to being sent to a Siberian gulag, included urbanites. In the UK House of Lords, Lord Segal asked his fellow peers to consider the plight of "countless thousands uprooted from their urban homes and dumped into bare, rural areas without adequate subsistence and little hope of survival" (Hansard HL Deb., 14 February 1979). Despite their lack of experience in farming, these internal relocatees were expected to undo the ecological destruction of the war years and make southern Vietnam agriculturally productive once again.

UK MP Stan Newens linked environmental destruction to a reduced standard of living, which was driving the boat people exodus:

"Should we not recognise that Vietnam has suffered an unparalleled measure of destruction, loss of life and misery in the wars of the past 30 or 40 years, and that that has been aggravated by natural disasters? Is it therefore not natural that many people wish to leave the country merely because of the lowering of the standard of living, as many of them have indicated?" (Hansard HC Deb., 18 June 1979)

His suggestion that wartime destruction and environmental degradation were driving the refugee crisis was struck down in the same debate by MP Ian Gilmour, who blamed the Vietnamese government for the exodus. Yet, since Vietnam's post-war policy of forced relocation to rural areas was prompted by ecocide, these arguments amount to the same conclusion: that ecocide was a root cause for their migration.

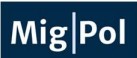

A confidential briefing to UK Prime Minister Margaret Thatcher from around the same time notes:

"[M]any of the ethnic Vietnamese of the former middle class in the South… who face the stark alternative of being transferred to a 'new Economic Zone' will prefer to risk leaving by sea" (Foreign and Commonwealth Office, 1979a:1)

This demonstrates acknowledgement within the British government that the policy of forced internal relocation and land reclamation, necessitated by the ecocide and exacerbated by extreme weather, was partly responsible for the ongoing refugee crisis. More broadly, the narratives of the day acknowledged that the boat people included both rural and urban dwellers fleeing for economic and environmental reasons, not only 'anti-Communist' sentiments.

Among the oral history participants, fear of the post-war 're-education' system and the NEZ relocation programme were stated reasons for fleeing Vietnam. In the California Irvine collection, eight interviewees mentioned the NEZ system. Additionally, 24 refugees discussed their experiences of forced agricultural labour in post-war Vietnam; in some cases, undertaken while they were children. Several interviewees in the British Library/Refugee Action collection, including former child labourers, describe relocation to NEZs and forced agricultural labour, including at re-education camps. These themes also arise in the Vietnamese Boat People Podcast (VBPP) and Voices of Vietnamese Boat People (VV) collections.

In addition to the NEZ system, recent estimates suggest 2.5 million people were interned in camps for supposed 're-education' (van Zyl, 2017). The archives of human rights research organisation Amnesty International provide an insight into the re-education system and its links to environmental conditions and the subsequent exodus. In 1979 Amnesty reported that, in addition to former military personnel, re-education camp detainees included "medical doctors, former civil servants and diplomats, journalists, academics, schoolteachers and writers" (Amnesty International, 1979). This challenges the narrative that all the detainees were former enemy combatants sent for re-education and punishment and suggests instead that anyone with potential ideological differences was considered suitable for forced agricultural labour. Some detainees remained in camps twenty years after reunification (Amnesty International, 1995). Throughout this period, the organisation noted that forced manual labour was a feature of the re-education system (Amnesty International, 1978 & 1987).

The reports of forced labour are relevant here since one of the main purposes – if not *the* main purpose – of the re-education camp system was the provision of vast amounts of slave labour to reconstruct the agricultural sector and transform the depleted landscape of southern Vietnam. A Foreign and Commonwealth Office (FCO) briefing contains testimony from the former Prime Minister of South Vietnam, Nguyen Van Loc, who was an inmate of a re-education camp between 1976 and 1980: "He described how inmates were forced to clear jungle swamps, farm the area and build a dam; two or three people died every week from malnutrition or disease." The briefing concludes: "Many of the [re-education] camps are simply pools of forced labour, and no attempt is made at 're-education' or indoctrination" (Foreign and Commonwealth Office, 1983).

Of the 158 oral histories analysed from the California Irvine collection, 18 participants, including children, had themselves been detained in a 're-education' camp, and 57 participants had seen at least one family member detained. These periods of internment ranged from a few months to 13 years, with some participants' family members never returning, presumed (or certified) dead. Several participants in the VBPP and VV collections had also been detained or seen family members detained. Many refugees confirmed that the purpose of these camps was to extract forced agricultural labour:

"[I]n reality, the re-education camp is more like the hard labor camp." (CI033, young adult male, former camp detainee)

"The job was farming, cultivating crops" (CI034, adult male, former camp detainee)

"[My father] had to work intense physical labor, farming for roots and vegetables" (CI043, adult female, daughter of camp detainee)

"[W]e cleared the forests with our hands.... After finishing planting, we moved to another location to prepare the soil for planting" (CI096, adult female, detained in a camp for five years)

"[W]e have to labor on the field, lao dong [labour], on the forest.... We have to grow up the vegetables" (CI002, male, age unspecified, former camp detainee)

"They had work like ten hours a day… they work on the farm, work on the mountain." (CI136, adult female, sister of two camp detainees)

Such testimony demonstrates that the re-education system was intrinsically linked to the wartime ecocide. It thus had the dual purpose of persecuting 'undesirable' groups while rehabilitating the decimated agricultural sector. Fear of internment in this brutal system was evidently a strong push-factor for people fleeing Vietnam.

In some cases, human rights violations including labour exploitation continued after detainees were released from these camps, as evidenced by five speakers in the VV oral history collection. VV011, an adult male former camp detainee, described how he was considered a "noncitizen… under strict surveillance" and his children were barred from further education. The stigma associated with past detention and forced labour thus also contributed to the decision to migrate overseas.

Child refugee VV010 described being left alone with his brothers when their father was detained in a re-education camp. The four children made a shelter from sticks and leaves, in which they lived for a year. When the youngest child was hospitalised with malnutrition, they were sent to an orphanage. When their father was released from the camp, the family was relocated to a NEZ:

"The people who drove us there told us to cut down the trees and build our own house…. We [the children] cleaned the yard, cut bamboo down, and planted tomatoes and corn, but we still did not have enough food. We would sometimes be so hungry that we would cut bamboo and boil the inside to eat. Sometimes we would get it out before it was done because we were so hungry. Sometimes we thought we would die." (VV010, male unaccompanied child)

When their father spoke up about conditions in the NEZ, he was jailed, leaving the children alone again: "If you tried to get out [of the NEZ], they would shoot you", VV010 explained. Nevertheless, the children escaped and, after many failed attempts, bribed an official and escaped by boat.

In a letter rebutting concerns raised by Amnesty International in 1980, Vietnamese officials confirmed that detainees released from re-education camps would be relocated with their families to NEZs to continue working the land (Socialist Republic of Vietnam, 1980). An FCO briefing noted that this forced resettlement "will almost certainly only be available in a rural area where life for them and their families is harsh and unfamiliar" (Foreign and Commonwealth Office, 1983). One former teacher described such an experience after he was released from three years' detention: "I had to become a rice farmer. I've never done something like manual labor" (CI067, adult male).

CI133, an adult male detained in a re-education camp for four years, similarly described being forced to grow rice for the state following his release from the camp; and being given only a starvation ration for himself. The speaker begins to describe life in the countryside, but the interviewer focuses their subsequent questions on the political situation, rather than CI113's experience of forced agricultural labour. This is one among many examples in the oral histories where the preconceptions of the interviewer about life in Communist-run Vietnam causes them to direct the participant to focus on the political context, to the potential exclusion of other topics. This choice is both informed by, and results in reinforcing, the dominant narrative that political opposition was refugees' sole reason for flight.

Forced agricultural labour, while prevalent in re-education camps, was not confined to that system. The cycle of imprisonment, forced labour and forced resettlement also applied to people caught attempting to leave the country (Amnesty International, 1981, 1982 & 1990). In the worst cases, those trying to escape, including children, were summarily executed or imprisoned for up to 12 years (Amnesty International, 1990). Those imprisoned were sometimes forced to do agricultural work or sent to a NEZ. In such cases, children could be separated from their parents. Child refugee CI083 was 13 years old when her family was captured while trying to leave Vietnam. Imprisoned for a month, she was forced to do manual labour: "There was nothing to eat and they forced you to work".

In other cases, families were forced to labour on farms, either full-time or in addition to their regular jobs or studies. CI053 was a child agricultural labourer: "my family went to the country [to] work on the farms. But not live there. We went to the farm in the morning and got back at night."

Some child refugees described how forced agricultural labour became a part of their post-war school curriculum:

"[T]hey began to call for youths to go do labour work, do irrigation projects, dig a ditch or a dam for the state. We were fed but not quite enough.... I got sick.... I was nothing but skin and bones" (BL026, female child labourer)

"[School children] study ½ of the time, the other ½ time work in the fields, planting sweet potatoes, yam, and sugar cane" (BL009, male child labourer)

"[K]ids like us we had to go on to these coffee plantations. We had to pick up the coffee beans… [and] turn these in to our school, and they would sell it" (CI059, male child labourer)

These children were directly exploited by the state to help reverse ecocidal destruction. In each case their experiences form part of their justification for leaving Vietnam.

Post-war food shortages and rationing were common themes in the oral histories, particularly for former child refugees. Some directly attributed this to the ecocide.

"[C]itizens were forced to become farmers and manual labourers to help rebuild the country. Food was scarce and rationed." (VB005, female child)

"[M]any families only had meat a few times a year.... You couldn't get meat even if you had money." (BL027, male, age not stated)

Violations of the right to food were gravest for those forcibly relocated to NEZs, where "the land allocated was often unproductive, the people relocated not skilled at farming and with no inclination to learn, and the tools they had were very primitive" (Dalglish, 1989:23). This is supported by oral history narrators, who recall the lack of preparedness for those forcibly relocated:

"[S]uddenly one night… [government officials] took the truck come over and load… [my neighbours] in and they took them away. They drive them to the countryside and they threw them in the forest" (CI076, adult female)

"They just came to [my cousin's] house and say 'okay you have 30 minutes to take whatever you can and get out the house' and they put them into a truck and load them into a farm" (CI063, female child)

"There were a lot of forest and infertile ground where they relocated us" (CI075, adult female)

Child refugee VB020, originally from a wealthy urban family, recalled being forcibly relocated to the Central Highlands, where his family struggled to sustain themselves. For others, the mere threat of being sent to a NEZ drove them to risk their lives as boat people:

"[M]y family was about to be sent to away [to a NEZ]… but the countryside wasn't ready for us, they were just going to throw us on some [uncultivated] land, how are going to survive?" (CI042, adult male)

"I have 8 children and my children still young so if we go to countryside we afraid they not survive" (CI076, adult female, husband detained in re-education camp)

"[T]hey keep making us work for the government [growing rice] but we don't get any money to survive" (CI133, adult male)

Each of the speakers quoted above, and others in the oral history archives, cited these experiences as part of their motivation for leaving Vietnam, frequently invoking the concept of survival. Their stories describe internal displacement and instability, brought about directly or indirectly by ecocide. In its attempt to reverse the effects of ecocide, the post-war government created policies that resulted in mass persecution of sections of the population; including IDPs and groups perceived to be ideologically opposed to the government's objectives. Each of the speakers above subsequently fled Vietnam, carrying with them their memories of forced labour, imprisonment, hunger and loss. Thus, their status as refugees was complete, since these individuals now found themselves, as the Refugee Convention puts it "outside the country of [their] nationality" (UN, 1954: Article 1(a)(2)).

## 4.4 Legal status of the refugees

The sections above make the argument that those fleeing for environmental reasons should have been considered refugees under the Refugee Convention since (a) they were subjected to ecocide and/or resultant agricultural, relocation or forced labour policies that constituted state *persecution*, (b) they were targeted for their *political opinion* or as members of a particular *social group*, and (c) they had *crossed an international border* in seeking asylum abroad.

Yet, in practice, this designation did not matter for the majority of those leaving Vietnam between 1975 and the mid-1980s. During this period, the Cold War rhetoric of Communist persecution of political opponents was more advantageous and compelling to Western decision-makers than the evidence of more complex drivers of the crisis, and most of those fleeing Vietnam before 1985 were given refugee status in third countries simply because they were perceived to be anti-Communist. From the late 1980s, when public opinion started to turn against resettling more Vietnamese refugees, Western governments began labelling the boat people as economic migrants to evade their legal responsibilities towards them. This resulted, from 1991 onwards, in large-scale repatriations of boat people to Vietnam from Asian transit camps (Human Rights Watch, 1997).

While the stories recounted above provide a powerful insight into ecocide as a direct or indirect migration driver, archival records and oral histories point to other push factors too. These include what has been described as the deliberate trafficking of Vietnamese citizens out of the country by the Vietnamese government.

In a letter to Margaret Thatcher, Prime Minister Kriangsak Chomanan of Thailand described the boat people migration as a "human export" (Chomanan, 1979). Likewise, a telegram from the Governor of Hong Kong noted that the boat people were Vietnam's "single most profitable export commodity", considering the bribes paid for departure (MacLehose, 1979). In the House of Commons, Philip Goodhart MP listed several "social and economic pressures" to depart, including fear of re-education camps and the "arduous, primitive and bleak" NEZ system, religious persecution and ethnic discrimination (Hansard HC Deb., 15 December 1978).

Minutes from a meeting between the FCO and UNHCR record how the UN agency was preparing to directly assist Vietnam in removing tens of thousands of "their unwanted population" from the country (Foreign and Commonwealth Office, 1979b). With the backing of some Western nations, on 30

May 1979 UNHCR signed an agreement to support the Vietnamese government with the expulsion through the ODP of those deemed undesirable by the state:

"The selection of those people authorized to go abroad under this programme will, wherever possible, be made on the basis of the lists prepared by the Vietnamese Government and the lists prepared by the receiving countries" (UNHCR, 1979).

The removal of these people from Vietnam with UNHCR's assistance was considered preferable to leaving them at the mercy of the sea. While the aims of the ODP were ostensibly family reunification and humanitarian ends, the criteria for one's name appearing on a government list, and the voluntariness of departure, remain unclear. An archived UK Government document contends that the intention of the ODP was to help Vietnam rid itself "of large numbers of ethnic Chinese" citizens (JCRV, 1980). Between 1979 and the mid-1990s, 650,000 people left Vietnam via the ODP.

The Vietnamese authorities profited massively from those desperate to leave, often extorting huge amounts of money to permit departure, even by boat. Dalglish (1989) notes how the ethnic Chinese population "were welcome to leave as long as they could pay" and that "local government officials would assist their journey" (pp.21 & 24).

Some members of this persecuted group, however, also cited reasons linked to the environment for their departure. One ethnic Chinese refugee, VV002, describes paying "$6,000 in gold so that my son and I could escape" from Vietnam. Their departure was hastened by the threat of being sent to a NEZ: "These were places with poor soil, little food, and no medicine. To avoid this, my family went underground, just like so many others did in the same situation". Several sources (Edholm et al., 1983; Chong, 1999; Lipman, 2020) suggest that the policy of deporting the Chinese population led Vietnamese people to fake Chinese identities in order to leave more easily. Seven oral histories from people bound for the USA and UK described pretending to be Chinese, including children.

These cases are significant since a 1982 study of the Vietnamese in Britain states that "the largest proportion were ethnic Chinese refugees" (Jones, 1982:15). Yet, slightly later archival documents (Edholm et al., 1983; Mougne, 1985) demonstrate that most refugees in Britain, particularly unaccompanied minors, were ethnically Vietnamese, with working class or peasant backgrounds, from rural villages or coastal fishing towns. Oral histories suggest that these were the demographics most likely to have been affected by ecocide. "Very few (4 per cent) of [the 100 families interviewed] said that they left because they hated communism", according to Edholm et al. (1983:36). (The very wording "hated communism" gives an insight into the bias of the questioning in Edholm et al.'s study.) Yet refugees' motivations for migrating may have been distorted in the official narrative by their adoption of fake Chinese identities, creating a lasting impression that Britain's Vietnamese population were predominantly Chinese merchants escaping political persecution, rather than poorer rural people or IDPs fleeing the long-lasting impacts of war, including ecocide.

Refugees interviewed for the British Library/Refugee Action oral history project gave a variety of reasons for leaving Vietnam, ranging from the fear of forced labour and internal relocation to political persecution.

Among the California Irvine oral histories, reasons for fleeing were equally diverse, and included fear of being drafted into the military; escaping from the Communists; hunger and poverty; lack of employment and education; confiscation of property; forced dispersal and forced labour under the NEZ and re-education systems; and the vague but oft-cited notion of seeking "freedom". Some children and young adults mentioned forced agricultural labour as a push factor:

"[W]hen you reach 17 years old you got to go to farm and work very hard… and no food support for you. A lot of people cannot survive and they die over there. So that's why it make us scared, so everybody want at that time… to escape out of Vietnam and find a free country" (CI032, male, left Vietnam aged 18)

"They may take all your family put in the mountain, take all your house, and everything... that's why I think most people want[ed] to escape Vietnam" (CI087, male unaccompanied minor)

Other former child refugees mentioned conscription and direct political persecution as motivations for fleeing:

"Because I was 13, my brother was 16... we would most likely have to go to the military. We would be fighting. So I knew that we had to leave." (CI010, male unaccompanied minor)

"[M]y father was... part of the old government meaning that none of the kids in the family would be able to go to college." (CI065, male child)

In the VBPP interviews, former child refugees cited the following push-factors: poverty, loss of family property or business, internal displacement caused by the conflict, re-education camps, child labour, military conscription, lack of medical care, and generalised fear, trauma and adversity. In the VV collection, reasons for leaving included forced agricultural labour, poverty, lack of medical care, family separation, religious persecution, lack of education, loss of family livelihood, state surveillance, military conscription and generalised fear and uncertainty.

Drivers for leaving Vietnam were therefore varied. Among them, issues related to land, agriculture and the broader environment played a significant and previously overlooked role in triggering the exodus, alongside other political, social and economic factors. Interviewees often had multiple reasons for fleeing, of which environmentally related drivers could be one aspect. Even where environmental factors are not explicitly mentioned, archival sources show that ecocide and extreme weather acted as first-order drivers, significantly influencing post-war government policy and accelerating the economic downturn, which pushed people to leave Vietnam to escape forced labour and starvation. The evidence shows that other second- and third-order impacts of ecocide included lack of access to education, family separation, poverty and uncertainty about the future; all reasons given by interviewees for leaving Vietnam.

# 5. Conclusion

The Vietnamese boat people exodus is an example of what we now call a mixed-migration flow. Despite the dominant narratives of our age, and the Cold War discourses of the time, the boat people were not solely 'anti-Communist' political refugees. They were fleeing for a range of reasons, including poverty, repression and conscription, as well as the direct impacts of ecocide and resultant government policies of forced relocation and forced labour.

As set out above, each of these drivers was the result of state persecution of particular political or social groups. At the time of their flight, some refugees had already faced persecution in the form of food denial, targeting with ecocidal weapons, forced internal displacement, forced labour, and destruction of the lands on which they relied. Others feared future persecution in the form of starvation, forced labour or forced displacement to fulfil state agricultural aims.

Thus, this paper concludes that those boat people fleeing for environmentally related reasons met the Refugee Convention definition due to being "outside the country of [their] nationality", "owing to well-founded fear of being persecuted for reasons of race, religion, nationality, membership of a particular social group or political opinion" (UN, 1954: Article 1(a)(2)).

Why does any of this matter now? It is more than 25 years since the last boat people were either resettled in third countries as refugees, or repatriated to Vietnam having failed the 'persecution test'. And yet, the problem of environmental degradation in southern Vietnam did not end in 1995. Like many regions of the world, Vietnam faces a range of environmental risks today, including from climate change, which may be contributing to new migration flows.

The exact mechanisms by which climate change drives migration are unclear (Owen & Wesselbaum, 2020). This lack of knowledge led the UN's Intergovernmental Panel on Climate Change (IPCC) to claim that "migration patterns, in the near-term will be driven by socioeconomic conditions and governance more than by climate change" (IPCC, 2022b:13). Yet the evidence above shows that socioeconomic conditions, governance and the environment are often interlinked. Meanwhile, other studies and organizations have found, or predict, strong correlations between human migration and climate change or other environmental degradation (e.g., IOM, 2014; IDMC, 2019; Moore & Wesselbaum, 2022).

Climate-related impacts, along with other forms of environmental degradation and ecocide, are increasingly being seen as violations of the right to a healthy environment when they affect human populations. They also violate a fast array of other human rights, including the rights to life, health and an adequate standard of living, among others. This paper argues that these phenomena are forms of persecution when they result from state policy and affect certain groups, communities or populations more than others. Often this persecution is 'passive' – for example when states or businesses externalise pollution, turning a blind eye to the suffering of marginalised populations in other jurisdictions. In other cases, the persecution is 'active' – as was the USA's deliberate destruction of the crops of rural peasant communities in South Vietnam. Whether active or passive, the states and businesses in question remain responsible for their actions or lack of action. When a state fails to protect, respect and fulfil the rights of rights-holders, whether in relation to harms caused by its own polices or from the (in)action of other states or business entities, the individuals affected may find their conditions becoming intolerable to the point that they choose to leave the state in which their rights are being violated. The moment they cross an international border, these individuals should have a claim to asylum under the 1951 Refugee Convention, so long as they can link the violations to their membership of a particular group.

Others have made coherent legal arguments for and against the expansion of the Refugee Convention definition to explicitly cover 'climate refugees'. This paper takes no position on that debate. Rather, the secondary aim of this paper has been to show that the existing definition of persecution in the Refugee Convention can encompass persecution committed via, or in the context of, ecocide. The definition can evolve without changing the text; as Kent & Behrman (2018) note: "the assertion that there is a singular concept of the refugee in law is simply wrong" (p.46).

Finally, the narrow view of 'persecution' as a violation of civil and political rights is expanding as our understanding of environmental crimes develops. Environmental harm violates a range of civil, political, economic, social and cultural rights. The effects on the victims – including hunger, poverty, disease and death – are no less grave than those forms of persecution more traditionally identifiable as bases for asylum – harassment, arbitrary arrest, political imprisonment and extrajudicial execution. Our reading of 'persecution' must therefore expand to include the environmentally linked violations that drive migration.

## Endnotes

1. This study uses the UN definition of a child as anyone under the age of 18 years.
2. Not all of the more than 1 million refugees who resettled in the USA were boat people; many arrived through direct resettlement programmes such as the Orderly Departure Programme and the 1988 Amerasian Homecoming Act.
3. The descriptors of "child" or "adult" describe the age of the person at the moment of departure from Vietnam, *not* their age at the time of the interview.

## Acknowledgements

**Funding information.** Funding for manuscript development was provided by the University of Amsterdam and the Migration Politics Fellowship.

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

\*\*\*\*\*