# Peer review of "Political or environmental refugees? Re-examining the flight of the Vietnamese boat people, 1975-1995"

_Migration Politics_

## Round 1 · Referee Report · Tamsin Barber (Referee 1) · 2025-1-27

Strengths

  1. This article provides a fascinating, novel and important insight into the previously established cannon on main causes for the flight of the Vietnamese boat people in the period after 1975.

  2. It provide a novel and synergetic link between different research areas.

  3. It is written in a clear and intelligible way, free of unnecessary jargon, ambiguities and misrepresentations

  4. It provides citations to relevant literature in a way that is as representative and as complete as possible

Weaknesses

1. The article needs to define its main contributions more clearly and these need to be better represented in article title and abstract and introduction. Specifically, the focus on ecocide and the narratives of child refugees should be foregrounded and their role and focus in the paper needs to be explained more clearly.

2. The role of ecocide is both over-stated and understated in different places in the article.

3. The article remains a bit conceptually weak. The article could better engage with the conceptualisations around ecocide .

4. The use of data should be be stronger and include greater analysis/interpretation of interview extracts

5. The presentation of data sources and participant characteristics needs to be clearer

Report

This article provides a fascinating, novel and important insight into the previously established cannon on main causes for the flight of the Vietnamese boat people in the period after 1975. While in the existing literature, an acknowledgement of other accompanying drivers does exist, this article brings these together in a more focused and sustained analysis. Importantly, it introduces ecological factors in a thoroughgoing way, that has not hitherto been explicitly acknowledged by most mainstream literature on Vietnamese refugee experience. The article takes an ethical approach to reanalysing existing data to reinterpret the past. In this way, it has merits in avoiding the retraumatisation of refugee communities. The article is generally well written and uses appropriate literature. It is written in a clear and intelligible way, free of unnecessary jargon, ambiguities and misrepresentations. It provides citations to relevant literature in a way that is as representative and as complete as possible. The article provides an important contribution for future work on the role of climate change in displacement and forced migration experiences.

Weaknesses
The article needs to define its main contributions more clearly and these need to be better represented in article title and abstract and introduction. Specifically, the focus on ecocide and the narratives of child refugees should be foregrounded and their role and focus in the paper needs to be explained more clearly.
In the introduction (section 1.1) the article puts forward that it aims to ‘frame the exodus in more nuanced terms as a mixed-migration flow with multiple drivers’, I like this framing and it seems to represent an accurate analysis. However, in other parts of the article the focus on climate change drivers and ecocide appears to be put forward as the main driver, this is a bit confusing for the reader. The author needs to decide where these claims fit within the overall argument. I suggest these sections need nuancing and putting into context with the other multiple drivers of exodus (including political, ethnic discrimination and harassment, hunger, poverty etc). There also needs to be a clearer statement of whether ecocide is seen as the primary cause of poverty and hunger or a parallel one.
The role of ecocide is both over-stated and understated in different places in the article. I suggest you make an argument for it being an important driver that intersects with others, but that it should be inserted into our understanding and evaluation of the intersecting drivers of the exodus of ‘boat people’ going forward. Connected to this, the conceptualisation of ecoide needs to be stronger. You should reflect critically on how useful notions of ecoide are and how they assist your analysis or otherwise. Otherwise the paper remains under theorised.
The focus on the accounts of child refugees in highlighted in section 2 introduces a fascinating element of the research. A justification for using these is given (i.e. in order to capture greater authenticity of accounts as children’s accounts were less likely to have become adulterated due to their likelihood to not be asked to retell their story as much as the stories of adults), however, there were are a few problems with your treatment of this issue. Firstly, this claim needs to be supported with a reference. Secondly, this issue needs greater exploration and should be reflected upon more in your interpretation of their narratives (how and why do they feature differently to the adult stories). Thirdly, other limitations of child narratives should be acknowledged – ie that the accounts of their experiences could be subject to other forms of social desirability effect or be less complete than the adult narratives (given perhaps their range of experience and/or interpretation frames). The fourth issue relates to a broader issue of the presentation of the data. While the oral history sources are listed in the methodology section, they need to be presented in the main section (after the extracts) in a more accessible way for the reader. It is difficult for the reader to remember what code relates to which data set. I suggest naming the data set, country, and where possible social characteristics of the speaker (including age, gender, origin). Sometimes, child/adult status are given but not consistently, and giving the age or age range would help to clarify the data and the flow of your argument.
In your main findings section, in addition to more clearly distinguishing between the claims made from the child and adult refugee data, you need to more clearly set out how often references to climate and ecocide appeared and how strongly they were emphasised in comparison to other factors. You do acknowledge that these issues were not part of the interview questions/schedule, but an overview of their occurrence and distinctiveness in the data (especially given they were not probed for by interviewers) would help the reader and give your argument a robustness.
Lastly, your data warrants greater interpretation. Often you present interview extracts without drawing out relevant aspects held within them.

There are some areas which need attention:
The same quote by Freeman and Huu (2003) is used twice in the article. Suggest finding another one to avoid repeating and diversifying your sources.
At the top of p 15 you have a very short descriptive stand-alone paragraph about the NEZs which needs developing further with some illustrations.
On page 17 you have a long list of data extracts that need more analysis and exploration

Under section 2.3 it would be worth reiterating the role of receiving country politics in prioritising representations of refugees as politically persecuted rather than fleeing starvation and climate disaster as a way to ‘sell’ hosting refugees in the West.

Requested changes

1. Define its main contributions more clearly and these need to be better represented in article title and abstract and introduction.
2. foreground their role and focus (of ecocide and child refugee narratives) in the paper.
3. Overall ensure the argument is developed consistently through the paper
4. Conceptualise ecocide more strongly and say how this intersects with other drivers
5. Present and analyse data more carefully
6. Pay attention to repeat use of quotes

Recommendation

Ask for major revision

---

## Round 1 · Referee Report · Anonymous (Referee 2) · 2025-2-6

Strengths

1, Well-written and organized with clear stakes and purpose. 2. Originality and solid presentation of facts and literature review. 3. The data collection is simply impressive. 4. author has done an admirable job of making this work accessible and convincing.

Weaknesses

The article spotlights child migrants, but the mention of child migrants is missing from the abstract and introduction. Needs to forefront this focus early on. It should not take 6 pages for us to get to this special theme, which the author is saying they are emphasising. I would recommend the author possibly thinking about working the children angle into the main argument, since it was such a driver in their postwar experience and migration. The key word ecocide can also be placed well in the abstract so it is given the same prominence it has in the article.

I am curious about the semantics for this first subpoint: The sections above make the argument that those fleeing for environmental reasons should have been considered refugees under the Refugee Convention since (a) they were subjected to ecocide and/or resultant agricultural, relocation or forced labour policies that constituted state persecution

Given that the Refugee Convention does not consider environmental factors, there might be confusion to the reader of the argument you want to make and the one that can be considered under the RC. You can separate this and make this clearer instead of lumping the persecution factor with your advocacy for environmental conditions.

Report

I agree with everything reviewer #1 said about this manuscript so will not repeat them. I think the publication fits the aims of the journal it is seeking to publish in. A great contribution to the field.

Recommendation

Ask for minor revision

---

## Round 1 · Referee Report · Rebecca Hamlin (Referee 3) · 2025-3-4

Strengths

  1. Very well written
  2. The authors is clearly extremely knowledgeable about the Vietnam War
  3. The article provides a novel interpretation about Vietnamese boat people
  4. Rich data collection - oral history and archival

Weaknesses

  1. Seems to go back and forth between two incommensurate arguments and needs to choose one and defend it with more grounding in the relevant literature.
  2. Many quotes from oral histories left without much discussion or analysis
  3. Emphasis on children is not convincing

Report

This is a very interesting and well written piece. I feel like there are two different directions this article could go, one of which would be an easier lift, and one of which would be more challenging.

The lighter lift would be to write an article that could draw more on the literature about how migration motives are inherently mixed (like Crawley and Skeplaris 2018), building on stuff about how the migrant/refugee binary is an over-statement (Hamlin 2021) and how even one of the most prototypical groups of refugees (Vietnamese boat people – see Tran’s “We are the Real, Original Refugees” 2022), are actually much more complicated than that, because political, economic, and environmental factors have always combined to make migration a messy thing, which is why the stories in these oral histories reflect that and are not clear-cut stories of political persecution. This is a contribution to the literature because previous accounts have not acknowledged this about Vietnamese boat people, even as they have been willing to consider this about other groups of border crossers. Then, the piece can theorize more on the reluctance, is it because of a narrow view of what forced migration is? Is it because of a reluctance to pinpoint US culpability? Most likely, a bit of both.

The heavier lift would be to actually argue that Vietnamese boat people were refugees in a different way than has been previously theorized, that they were refugees under the 1951 Refugee Convention because they were the targets of ecocide. This framing would require a sharper definition of ecocide, and a much deeper engagement with the legal literature on bringing environmental factors into the Refugee Definition. This would be a more doctrinal contribution.

Either one of these articles would be interesting and a contribution. But, they are somewhat mutually exclusive. And as it currently stands, the article seems to be arguing in one frame at times, and then shifts into the second frame at other moments. So, my biggest piece of feedback is that I think the author needs to choose between these contributions and stay in that lane more definitively.

Either way, I must confess that I’m not sold on the child aspect of the piece the way it stands now, and find it to be something of a distraction from the larger argument. First, it is not signaled from the beginning, comes as a surprise on pg 6. But more importantly, it’s not exactly clear how the particular story of children fits with the larger argument. It’s not clear to me that children’s accounts can give insights into why a family decided to migrate. I imagine many parents would not tell thier children all the details of their decision-making and would try to shield them as much as possible from the worst aspects of thier suffering.

So, if the author is going with option 1, I would downplay the child as key informant angle, and just say that many of the boat people were children who were clearly motivated by a range of factors, which were unlikely to be political. If the author goes with option 2, a much stronger case would need to be made that children were the deliberate targets of ecocide. As it stands now, I was convinced that they were incidental and unfortunate targets, but not more.

Finally, I strongly agree with the reviewer who said that the quotes from oral histories need to be explicated more and can’t just stand alone.

Requested changes

  1. Choose one focus for the main argument/contribution and stick to it
  2. Needs to explicate the oral history quotes
  3. Needs to justify the emphasis on children more

Recommendation

Ask for major revision

---

## Round 2 · List of Changes

Response to reviewers’ reports on Vietnam paper

Ecocide

The hitherto undocumented role of ecocide in some boat peoples’ migration remains the central argument of my paper. However, to give the paper contemporary (rather than merely historical) relevance (something that is praised by Report 1 (R1)), I show how environmentally related persecution exists and should be considered to be just as important as other forms of persecution under the 1951 Refugee Convention.

R2 states that “ the Refugee Convention does not consider environmental factors”. This is not quite correct. The Refugee Convention neither rules in nor out environmental factors as a cause/outcome of persecution. I tackle this head-on in the Introduction (section 1.1), to which I have also added extra explanatory material.

My article shows that ecocide can both be a form of persecution (in this case by South Vietnam and the USA) and also form the conditions in which persecution is perpetrated (in this case by the post-war reunification government of Vietnam).

Thus, I have maintained the argument that ecocide is a form of persecution and is thus a contributing factor in eligibility for refugee status for certain groups. I have also added this focus to the Abstract.

I have addressed comments by all three reviewers that ecocide needs more conceptual grounding in the article and should be addressed throughout, including in the Abstract. I have added a new section on “Ecocide” to section 1.2 - scope and terminology, to further elaborate on the usefulness of the concept of ecocide to my argument. I have also outlined my argument in relation to legal counterarguments in section 1.1. Given the word count restrictions, these new sections are short and elsewhere in the article this is a case of being more precise with the language of ecocide rather than providing further, lengthy theoretical discussion of the term.

In the Conclusion, I have added a short paragraph on the evolution of the definition of persecution to include ecocide, and explained that this paper is not an attempt to rewrite the Refugee Convention or provide lengthy legal arguments, but rather the presentation of a relevant case study to encourage an evolving approach to the Convention’s implementation.

Child migrants

All three reviewers raise questions on the validity of the focus on child migrants. This focus was retained after detailed discussion during the residency process with numerous peers and other reviewers, in which I explained the unique importance of children’s voices in migration studies.

To respond to reviewers’ confusion on this topic, however, I have tried to provide more information in the article as to the importance of this focus. This includes new text reminding the reader of the child-centred approach throughout the paper, and additional literature cited in section 2.2.2 to back up my claims. I have also foregrounded the issue of children’s testimonies in the Abstract and Introduction so that it doesn’t come as such a surprise in the methodology section.

In particular, I have added text to explain that children made up some 50% of the boat people. This should help to explain why I have tried to include a near-representative sample (approx. 40%) of children’s narratives in my study. It is not that my study focuses entirely on children’s voices, but that they are given equal space to express their individual perspectives and experiences (unlike in many other studies). I have also tried to show more clearly when I am relying on children’s voices or adult voices, by adding demographic information after each of the oral history quotes.

Data presentation and analysis

As requested by R1, I have provided more analysis of the oral history quotes where this was lacking in the Findings (section 4), and I have standardised the way they are presented in the article, including with more information about the individual speakers.

As suggested by R1, I have provided more analysis of how often the “occurrence and distinctiveness in the data” of issues connected to ecocide appear in the oral histories (see sections 4.2 and 4.3).

In the response to R1: I have deleted one reference to the Freeman & Huu quote that appeared twice, and I have provided more background information on NEZs in section 2.2.1.

To mitigate the additional words added by these changes, I have removed the paragraph from section 1.2 that described the current threats of climate change to the Mekong Delta region, and made a few other cuts elsewhere in the document.

I thank the reviewers and editor for their helpful feedback and hope these changes will suffice to render the paper acceptable for publication.

Saphia Fleury, 13 March 2025

---

## Editorial Decision

accepted_in_target_journal